*Report*

EMBO
Molecular Medicine

# Defining the optimal animal model for translational research using gene set enrichment analysis

Christopher Weidner[1], Matthias Steinfath[1], Elisa Opitz[1], Michael Oelgeschläger[1] &
Gilbert Schönfelder[1,2,*]

## Abstract

The mouse is the main model organism used to study the functions of human genes because most biological processes in the mouse are highly conserved in humans. Recent reports that compared identical transcriptomic datasets of human inflammatory diseases with datasets from mouse models using traditional gene-to-gene comparison techniques resulted in contradictory conclusions regarding the relevance of animal models for translational research. To reduce susceptibility to biased interpretation, *all* genes of interest for the biological question under investigation should be considered. Thus, standardized approaches for systematic data analysis are needed. We analyzed the same datasets using gene set enrichment analysis focusing on pathways assigned to inflammatory processes in either humans or mice. The analyses revealed a moderate overlap between all human and mouse datasets, with average positive and negative predictive values of 48 and 57% significant correlations. Subgroups of the septic mouse models (i.e., *Staphylococcus aureus* injection) correlated very well with most human studies. These findings support the applicability of targeted strategies to identify the optimal animal model and protocol to improve the success of translational research.

**Keywords** animal model; GSEA; inflammation; transcriptomics; translational research

**Subject Categories** Immunology; Systems Medicine

## Introduction

The mouse has been used as model for heredity analysis since the 19th century. The development of targeted genetic modifications in mice in the late 1980s and the finding that almost 99% of mouse genes have an equivalent in humans established mice as the main model organism for the analysis of the molecular mechanisms underlying human diseases.

However, recently, the relevance of mice as a valuable model organism for translational research has become increasingly controversial. This discussion has been renewed by the study of Seok *et al* (2013) that compared transcriptomic data from human inflammatory diseases with data from mice challenged with inflammatory stimuli and the work of Takao & Miyakawa (2015) that reanalyzed the same datasets using different strategies and algorithms.

Both groups strongly focused on classical gene-to-gene comparisons and examined the global congruency of the differentially expressed genes. The traditional strategy to analyze transcriptomics on a single-gene level is well justified for perturbations that have very large impacts on individual genes and for biomarker identification. However, these conventional approaches suffer from severe inefficiencies because they make use of only a minor fraction of the information contained in whole-genome transcriptomic datasets. A major limitation is the need to set an arbitrary threshold for the determination of differentially expressed genes (which is also a matter of discussion in Seok *et al*, 2013, and Takao & Miyakawa, 2015). This requirement generally restricts the analysis to a few highly up- and downregulated genes and completely disregards the information from the majority of the many thousands of transcripts detected in biological samples, thereby providing reasonable objections against biased selection. In contrast, gene set enrichment analysis (GSEA), which was established for use in metabolic studies in the 2000s (Subramanian *et al*, 2005), first maps all detected unfiltered transcripts to the intended pathways (or other "sets of interest") to allow the analysis of thousands of transcripts potentially involved in the concerted regulation of specific pathways. GSEAs also include moderately regulated (and nonregulated) genes that would otherwise be lost due to experimental background noise. Additionally, to identify regulated signaling pathways, a statistical evaluation is subsequently performed using running sum statistics. Importantly, this approach is fundamentally different from assigning Gene Ontology (GO) terms or pathways to genes *after* filtering for strongly regulated genes and avoids the problems associated with cutoff values. Using this strategy, we analyzed the human and murine datasets used by Seok *et al* (Table 1 in Seok *et al*, 2013) and focused on pathways that were specifically assigned to inflammatory processes. We identified and compared pathways that

1 Department of Experimental Toxicology and ZEBET, German Federal Institute for Risk Assessment (BfR), Berlin, Germany
2 Department of Clinical Pharmacology and Toxicology, Charité-Universitätsmedizin Berlin, Berlin, Germany
 *Corresponding author. Tel: +49 30 18412 2286; E-mail: gilbert.schoenfelder@bfr.bund.de

were regulated in either human or mouse studies and determined whether the regulation of inflammatory pathways in one model (e.g., human) could be predicted by a second model (e.g., mouse). Finally, this strategy facilitated the identification of the optimal mouse model for translational research of severe inflammatory disorders.

# Results

## Correlation of inflammatory pathway analysis: human–human, mouse–mouse, and human–mouse comparisons

First, we applied GSEA to identify the relevant signaling pathways. We included expression data for all transcripts, thereby avoiding biased filtering of a few highly regulated genes. Because mouse models aim to mimic a specific disorder using different experimental manipulations (i.e., severe inflammatory diseases), we specifically focused on pathways involved in immunological processes annotated by the BioCarta, Reactome, and Kyoto Encyclopedia of Genes and Genomes (KEGG) databases. Then, we compared the pathway regulation profiles between the human–human, mouse–mouse, and human–mouse datasets.

The human datasets correlated very well with one another, with average positive and negative predictive values of 61% (Fig 1A) that represented a 35% increase against overlap by chance (Fig EV1). A total of 96% of the human–human comparisons showed significant correlations in their pathway activities ($P < 0.05$). However, various distinct mouse datasets showed only slight intraspecies correlations in inflammatory pathway regulation, with average positive and negative predictive values of 44% (11% increase against overlap by chance). A total of 47% of the mouse–mouse correlations were statistically significant (Figs 1A and EV1). Moreover, both the congruency of pathway regulation within one species and the number of significantly correlating pathways within one species were species dependent ($P \leq 0.0001$, Figs 1A and EV1).

Strikingly, analysis of the overlap between all human and mouse datasets revealed average positive and negative predictive values of 48% (representing a 19% increase against overlap by chance), and 57% of human–mouse comparisons showed significant correlations ($P < 0.05$) (Figs 1A and EV1). These findings indicate that certain mouse models have the potential to mimic human inflammatory signaling processes much better than others.

## Identification of mouse models with high predictivity for human sepsis

The comparison of inflammatory pathway regulation demonstrated high diversity in the degrees of their correlation that could likely be attributed to the diverse etiology and/or treatment protocols underlying inflammatory diseases. As depicted in Fig 1B, a subgroup of mouse models correlated very well with most human studies, with average positive and negative predictive values of 62% and an average 32% increase against overlap by chance (dotted line). This result comprises infection models induced by *Staphylococcus aureus* injection (intravenous or intraperitoneal) or cecal ligation and puncture (CLP), in concordance with previous reports (Buras *et al*, 2005;

Fink, 2008). In contrast, studies based on intoxication by lipopolysaccharide (LPS) gavage or intratracheal injection of *Streptococcus pneumoniae* showed no correlation with the selected human septic models (Fig 1B), raising questions as to the applicability of these mouse models to mimic systemic inflammatory disorders (Buras *et al*, 2005; Fink, 2008).

## Identification of inflammatory pathways shared between human and mouse models

We identified the underlying inflammatory pathways that were collectively induced in the selected human and mouse models (Fig 1B, dotted line). The shared pathways included FcγR-mediated phagocytosis, Toll-like receptor pathways, interleukin-6 and interleukin-1 receptor pathways, and the complement cascade (Fig 2 and Appendix Fig S1). Strikingly, modulators of these signaling pathways have recently been translated into promising clinical trials against conditions such as psoriasis (Hennessy *et al*, 2010; Brennan, 2014), rheumatoid arthritis (Jones *et al*, 2010), and delayed graft functions (Hennessy *et al*, 2010), among others (Dinarello *et al*, 2012).

# Discussion

Animal models have long been used for determining disease mechanisms and drug discovery (Bolker, 2012). Low success in clinical trials, particularly in the field of sepsis therapy, led to skepticism regarding the predictivity of animal models (Cohen *et al*, 2015). Additionally, controversial discussions were provoked by the very different conclusions drawn from the same data after applying differing data analysis strategies (Seok *et al*, 2013; Shay *et al*, 2015; Takao & Miyakawa, 2015; Warren *et al*, 2015). Thus, robust and reliable bioinformatics techniques are needed for the analysis of complex "omics" data to systematically identify the optimal animal model for a given human disorder. Notably, an optimal model assessment contributes to animal welfare because it can help avoid overinterpretation and unnecessary repetition of animal experiments that might not correlate with the human situation.

One major critique of the data analysis strategy proposed by Takao & Miyakawa (2015), which concluded that "genomic responses in mouse models *greatly* mimic human inflammatory diseases", was the restriction to a few genes that were differentially expressed in *both* species (Warren *et al*, 2015). We extended this strategy to additional human and mouse datasets (Table 1 in Seok *et al*, 2013) that were not considered by Takao and Miyakawa. The correlation analysis confirmed significant correlations for most of the intra- and interspecies comparisons (121 out of 137 with $P < 0.05$, Appendix Fig S2). Apparently, the single-gene approach used by Takao and Miyakawa tended to overestimate the correlations between mouse and human data because only overlapping differentially expressed genes were compared to one another, and all relevant information from the other genes involved in inflammation was skipped. Consequently, this approach did not allow the identification of mouse models that better mimicked human disease. Notably, this data-handling strategy is not appropriate for multiple hypothesis correction because it will yield very few genes for some dataset correlations. These limitations and the discrepancy between

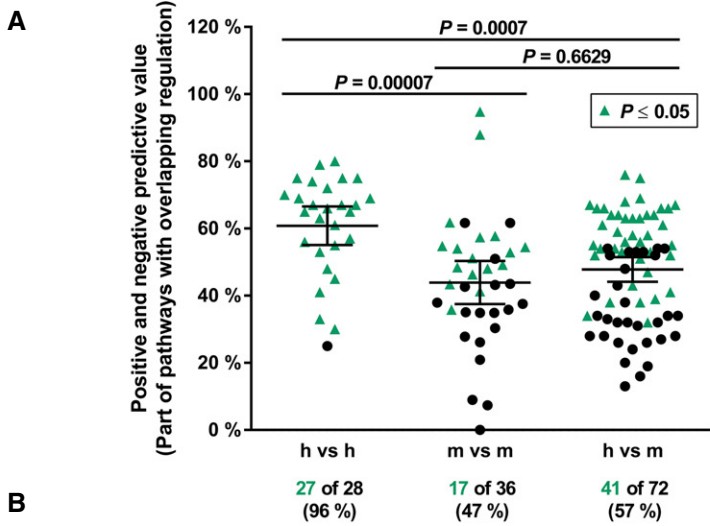

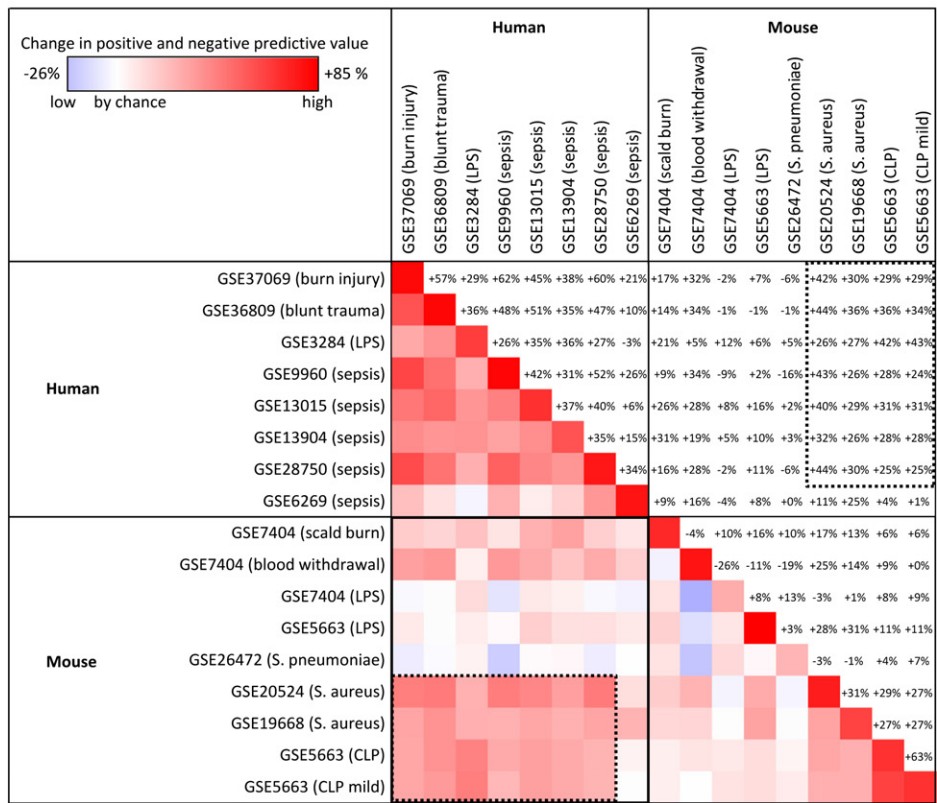

**Figure 1.  Identification of inflammatory mouse models that show high correlation with human diseases.**

The regulation of inflammatory pathways was assessed by gene set enrichment analysis (GSEA) using unfiltered gene expression data from 8 human and 9 mouse studies.

A   Significantly regulated pathways were compared between two datasets from human (h) and/or mouse (m) studies. The degree of pathway overlap is depicted as the mean predictive values between these two datasets. Studies that revealed pathway overlap significantly greater than that expected by chance ($P \leq 0.05$) are labeled with green triangles. The numbers of significantly correlated studies are given below the datasets. Lines indicate the mean $\pm$ 95% confidence interval. The $P$-values for each pair of datasets were calculated using a chi-squared test, and the $P$-values for the comparison of species effects were calculated using the Kruskal–Wallis test followed by Dunn's multiple comparisons test and Bonferroni correction.

B   Correlation matrix of pathway comparisons among human and mouse inflammatory studies. The overlap of pathway regulation is shown as the average change in the positive and negative predictive value over expectation by chance (blue, decrease, low correlation; red, increase, high correlation). The comparison of human with murine datasets revealed a subgroup of experimental murine models that were highly correlative to human clinical studies (dotted line), consisting of the *Staphylococcus aureus* injection and the cecal ligation and puncture (CLP) models. In contrast, lipopolysaccharide (LPS) gavage and intratracheal infection with *Streptococcus pneumoniae* showed no correlation to human inflammatory diseases.

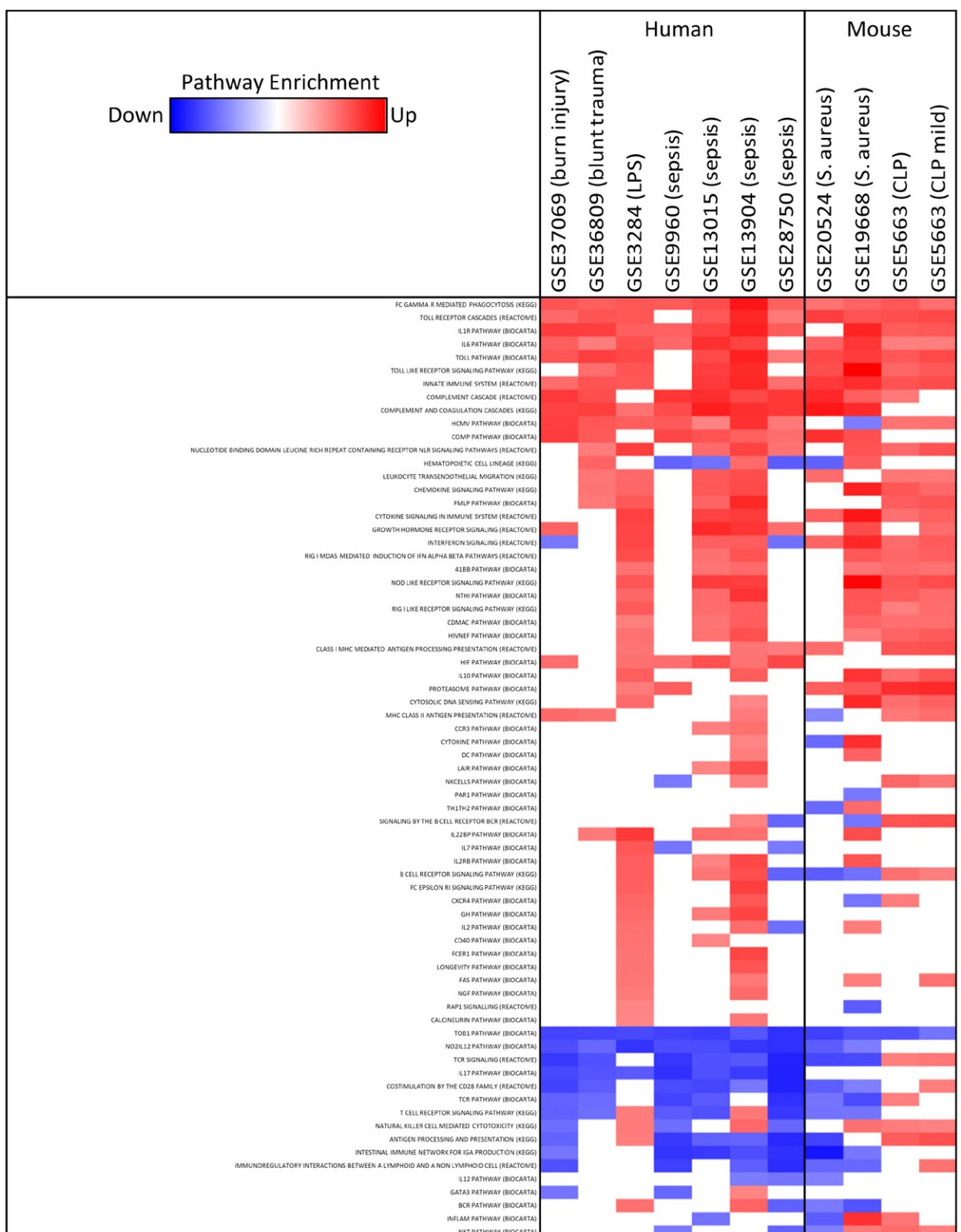

**Figure 2.    Identification of regulated signaling pathways shared between human inflammatory diseases and selected mouse models.**
The datasets were from the subgroup of experimental murine models that were highly correlative to human clinical studies. Regulation of inflammatory pathways
was determined by GSEA and is visualized according to the normalized enrichment score (blue, decreased expression, FDR ≤ 0.25; red, increased expression, FDR ≤ 0.25; white,
pathway not detectable or not changed, FDR > 0.25).

the opposite conclusions drawn by Seok *et al* (2013), who concluded that a poor correlation existed between humans and mice, disclose a strong need for more standardized and unbiased data analysis strategies.

Gene set enrichment analysis overcomes the aforementioned limitations of traditional gene-to-gene interpretations because it makes use of a "no-cutoff" strategy by including all experimental information, irrespective of the magnitude of the signal changes. Thus, GSEA avoids arbitrary factors during gene selection and includes information from the majority of the many thousands of transcripts detected in biological samples (da Huang *et al*, 2009). After assigning the detected transcripts to biological pathways, GSEA uses nonparametric Kolmogorov–Smirnov enrichment statistics to estimate the regulation of these pathways. These set enrichment approaches have been shown to be more robust for comparisons of transcriptomic data between different platforms or with different experimental models or clinical cohorts (Spinelli *et al*, 2015). Therefore, these approaches are currently recommended for data analysis for human risk assessment of chemicals and may lead to reductions in animal use in toxicological research (National Research Council (US), 2007; Pielaat *et al*, 2013; European Food Safety Authority, 2014). Using GSEA, we were able to identify particular mouse models (i.e., *Staphylococcus aureus* injection and the CLP model) that showed inflammatory pathway regulation profiles that were similar to the human situation; notably, most of these pathways are currently promising clinical targets in several inflammatory disorders. Nevertheless, from the clinical outcome, it is known that sepsis (in contrast to other inflammatory disorders) is a complex syndrome involving both *pro*-inflammatory (early phase) and *anti*-inflammatory (later phase) responses (Hotchkiss *et al*, 2009). Addressing this complex biphasic pathogenesis may require more than suppression of the pro-inflammatory processes. Instead, recent studies have indicated that inhibiting the sepsis-induced apoptosis of immune cells that caused lethal immunosuppression was the appropriate strategy for the treatment of severe sepsis (Hotchkiss & Opal, 2010; Fink & Warren, 2014). Taking the complex biphasic pathogenesis of sepsis into consideration, the recruitment of participants and the selection of adequate time points to define the septic model are of high importance. The same is true for the preclinical murine model.

We agree with Warren *et al* (2015) that drugs physically work at the molecular level (e.g., by binding to specific proteins). However, increasing evidence from genetic and molecular biology studies has shown that cellular processes and pathways are generally not affected by the alteration of a single gene or protein but instead are the result of alterations to a group of interacting proteins. This issue highlights the importance of analyzing signaling networks involved in diseases and their underlying biological processes (Yu *et al*, 2015). Consequently, the current pharmacological strategy is to modulate a specific pathway or parts of a pathway involved in immune processes, and pathway-focused strategies are valuable approaches for the identification of relevant therapeutic targets. The proceeding steps inevitably include the definition of possible key players in the relevant pathways. We were able to identify a list of genes involved in the Toll-like receptor cascade (Reactome) pathway (Table EV1) that were upregulated in at least 9 of 11 datasets where the human and mouse inflammatory studies were significantly correlated. For instance, congruently upregulated genes

included the CD14 molecule, the lipopolysaccharide-binding protein (LBP), and the Toll-like receptors (TLRs) 2, 4, 6, and 8. CD14 is an accessory receptor for TLRs, and ongoing clinical trials indicate that CD14 inhibition is a promising strategy to treat sepsis (Verbon *et al*, 2001; Egge *et al*, 2015). Furthermore, LBP is an acute-phase biomarker for sepsis (Schumann & Zweigner, 1999), and the TLR family was recently reported to be a promising target for the treatment of sepsis (Savva & Roger, 2013).

Seok *et al* (2013) and Takao & Miyakawa (2015) analyzed the treatment or disease effects at the genome-wide level to evaluate the capacity of mouse models to mimic human inflammatory diseases. Despite the given limitations (Shay *et al*, 2015), such as the differing cellular compositions (i.e., lymphoid to myeloid ratio) of humans and mice, and the treatment effects in humans that were ignored, the aim of both studies was to gain a deeper understanding of the conserved effects between humans and mice on the molecular level. Notably, most of these candidate genes are excluded from traditional gene-to-gene analyses due to their moderate expression changes. Therefore, the gene set enrichment analysis presented here offers a valuable tool for drug target identification.

Using GSEA, we were able to provide more detailed information. We can certainly conclude that biologically relevant *parts of* "genomic responses in mouse models mimic human inflammatory diseases" (Seok *et al*, 2013; Takao & Miyakawa, 2015) despite these limitations in the datasets. Our analysis supports the assumption that the identification of new pharmacological targets for the treatment of inflammatory diseases does not necessarily require the complete comparison of the whole transcriptomes of human samples with the transcriptomes from mouse models (as also discussed by Shay *et al*, 2015). Instead, it is likely imperative for pharmacological target identification to focus on a particular (patho)physiological process. Therefore, it is necessary (i) to identify regulated pathways and molecules that are exactly shared between the given human disorder and experimental mouse models and (ii) to identify the best mouse model for that human disorder (e.g., burn or sepsis). We emphasize that the GSEA approach will not overcome some basic difficulties in developing the appropriate animal model *before* any (transcript)omics data have been generated from this model. Generally, similar agents are used in both humans and mice to induce diseases with similar phenotypes. The same holds true for the introduction of genetic defects that target species homologs. This phenomenon does not reflect the possible variety of outcomes resulting from the diversity of biological processes across species barriers (i.e., mechanisms of adaptation, different pharmacokinetics upon gene deletion or treatment with similar triggers/inducers). Importantly, GSEA is not a tool to predict how to *a priori* design new model systems, but it is an effective tool to decide how to interpret existing data in a standardized manner, which might add value to the careful selection of the correct animal model and might avoid unnecessary and misleading translational studies.

Currently, 241 mouse strains are available at The Jackson Laboratories to investigate "infectious diseases". This number reflects only a small fraction of all inflammatory mouse models that are either experimentally induced or genetically modified. Inbred mouse strains that provide a basis for subsequent disease models are highly divergent in their immune response patterns as a result of genetic

mutations and polymorphisms (Sellers et al, 2012). Knowledge of these genetic profiles is essential for the accurate selection of disease models. Therefore, adequate model assessment is one of the most critical factors for success in translational research for the selection of either already existing disease models or the appropriate laboratory strain genetic background to facilitate further disease model development. We believe that the use of previously published datasets, ongoing discussions on experimental improvements (Shay et al, 2015), as well as targeted strategies for model assessment and the gene set enrichment analyses described here have great potential to systematically define the optimal mouse model and protocol for a given human disorder (and are not restricted to immunological studies). These resources should be increasingly used to improve the success of translational research and to reduce the number of animal studies. Indeed, the recent debate concerning failed clinical trials linked to inappropriate animal models demands a re-evaluation of the respective mouse models and the utilization of the growing availability of transcriptomics data and systems biology approaches.

The current discussion on different approaches and algorithms clearly discloses a general limitation of genomic studies: the bioinformatics analyses are not well standardized, and the resulting conclusions are prone to bias. If the same sets of experimental data can lead to diametrically opposite conclusions, then the relevance and outcome of genomic (and other "omics") studies for purposes other than explorative approaches are negligible and strongly require improvement. Thus, better standardization of statistical analyses and data interpretation are urgently needed and should be addressed to improve the use of omics data for regulatory purposes (National Research Council (US), 2007; Pielaat et al, 2013; European Food Safety Authority, 2014).

The novelty of our approach encompasses several key aspects. First, by using a GSEA approach instead of a single-gene analysis, it is possible to circumvent any problems associated with the subjective setting of gene expression thresholds and gene filtering, which led to opposite conclusions by Seok et al (2013) vs. Takao and Miyakawa (2015), and to enable analysis in an unbiased, standardized manner. Second, by using pathway-based GSEA, the focus is exclusively on genes annotated as involved in inflammatory processes, thereby specifically addressing the (patho)physiological process of the question. Third, our approach is able to identify (already existing) mouse models that have high predictivity for the human condition of interest, thereby reducing unnecessary animal studies while increasing success in translational research.

# Materials and Methods

### Datasets for human diseases and mouse models

The datasets used in this study were the same as those selected by Seok et al (2013). The transcriptomic data were downloaded from the gene expression omnibus (GEO) database and consisted of 8 human and 9 mouse studies. Accession numbers and details are listed in Appendix Table S1. If different time points were available, one was arbitrarily chosen as indicated in Appendix Table S1. GSE10474 (human ARDS) and GSE19030 (mouse ARDS) were excluded due to the lack of healthy controls.

### Gene set enrichment analysis (GSEA)

The GSEA approach (Subramanian et al, 2005) was used to systematically analyze the transcriptomic data modulation of inflammatory pathways. GSEA allows the detection of congruent changes in gene expression for a defined pathway and thus is sensitive to slight alterations in pathway regulation irrespective of the differential expression of particular genes. Disease/treatment groups were always compared to the corresponding control groups. Mouse gene symbols were collapsed to Human Genome Organization (HUGO) symbols in the GSEA tool. Inflammatory pathways (95 in total) were manually downloaded from the BioCarta, Reactome, and KEGG databases (Dataset EV1). GSEA was performed with the following parameters: 1,000 gene set permutations, weighted enrichment statistics, gene set size between 15 and 500, and signal-to-noise metrics. Regulated pathways were considered statistically significant if the false discovery rate (FDR) was $\leq 0.25$. The GSEA-derived normalized enrichment score was used for the visualization of pathway regulation.

### Comparison of pathway regulation

Building on the GSEA results, there were three possible outcomes for pathway regulation: (i) the pathway was significantly upregulated with an FDR $\leq 0.25$, (ii) the pathway was significantly downregulated with an FDR $\leq 0.25$, or (iii) the pathway was not changed (FDR $> 0.25$) or too few genes in that pathway were detectable. The regulation of inflammatory pathways was compared between two datasets (models) in each case by calculating the positive and negative predictive values. This approach was used to investigate whether the pathway regulation of model 1 could be predicted by model 2. Thus, the positive predictive value was estimated by the number of overlapping upregulated pathways in models 1 and 2 divided by the total number of pathways upregulated in model 2. Correspondingly, the negative predictive value was estimated by the number of overlapping downregulated pathways in models 1 and 2 divided by the total number of pathways downregulated in model 2.

Positive and negative predictive values were compared to the values expected by chance. For instance, if 10% of the pathways were upregulated in model 1 as defined above, the positive predictive value to model 2 would be 10% if the regulation processes in both models were independent from one another. This approach allows the estimation of the amount of information that can be gained from model 2 regarding model 1. A chi-squared test was performed to estimate the statistical significance of the prediction calculation. This test is based on a three by three contingency table, which represents all three possible outcomes of the pathway analysis. Thus, all inflammatory pathways and all genes belonging to these pathways were considered in the analysis. As a consequence, the analyses included all genes assigned to the inflammatory pathways, and no biased filtering of highly up- or downregulated genes was required.

To determine the statistically significant differences in intraspecific pathway correlations between humans and mice, the Kruskal–Wallis test followed by Dunn's multiple comparisons test and Bonferroni correction were performed using the R package FSA function. Furthermore, a two-sample test for the equality of

## The paper explained

### Problem

Mice represent a valuable model organism for translational research to study the functions of human genes. Although it is generally accepted that most biological processes in the mouse are highly conserved in humans, the relevance of the mouse as a model organism for translational research has become increasingly controversial. This discussion has been renewed by recent reports comparing the same transcriptomic datasets from human inflammatory diseases with data obtained from frequently used mouse models. Strikingly, the different research groups obtained contradictory conclusions using the same experimental data. For instance, whereas one group of researchers inferred that "genomic responses in mouse models *poorly* mimic human inflammatory diseases", another group concluded that "genomic responses in mouse models *greatly* mimic human inflammatory diseases". Thus, the current debate clearly discloses a general limitation of genomic studies: the bioinformatics analyses are not well standardized, and the resulting conclusions are prone to bias.

### Results

To overcome subjective gene filtering and ineffective gene-to-gene comparisons, we propose the use of robust approaches, such as gene set enrichment analysis (GSEA), that comprise all genes relevant for the biological question under investigation (e.g., inflammation), including the majority of only slightly regulated genes. Thus, we reanalyzed the same datasets that led to the current controversy and focused on pathways that were assigned to inflammatory processes in either humans or mice. In contrast to other studies, we included expression data from all inflammatory genes without the need to set arbitrary thresholds that would lead to informative losses of the majority of the gene expression data. Analyses between all human and mouse datasets revealed a moderate but significant overlap. However, a particular subgroup of septic mouse models (i.e., *Staphylococcus aureus* injection or cecal ligation and puncture) correlated very well with most human studies.

### Impact

These findings strongly support the applicability of targeted analysis strategies such as that presented here to define the optimal animal model and treatment protocol for a given human disorder. These approaches have the potential to improve the success of translational research and to reduce the number of animal studies required.

proportions with continuity correction was performed on the species-related number of significantly correlated pathways using the R function prop.test.

**Expanded View** for this article is available online.

## Acknowledgments

This work was financed by the German Federal Institute for Risk Assessment (BfR).

## Author contributions

CW, MS, and GS conceived the study. GS supervised the research. CW and MS conceived and designed the experiments. CW and MS performed the experiments. MS performed the statistical analyses. All authors analyzed the data and wrote the manuscript.

## Conflict of interest

The authors declared that they have no conflict of interest.

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
