## [Review Process File · EMBO Molecular Medicine]

Defining the optimal animal model for translational research using gene set enrichment analysis

Christopher Weidner, Matthias Steinfath, Elisa Opitz, Michael Oelgeschläger, Gilbert Schönfelder

Corresponding author: Gilbert Schönfelder, Federal Institute for Risk Assessment BfR

Review timeline:

Submission date:	02 November 2015
Editorial Decision:	25 January 2016
Revision received:	15 April 2016
Editorial Decision:	29 April 2016
Revision received:	10 May 2016
Accepted:	11 May 2016

Transaction Report:

Editor: Roberto Buccione

1st Editorial Decision

25 January 2016

Thank you for the submission of your manuscript to EMBO Molecular Medicine and many apologies due to intervening holiday season and the delay in retrieving the evaluation from one reviewer, who ultimately did not deliver.

We have now heard back from the three Reviewers whom we asked to evaluate your manuscript.

As you will see the Reviewers issues are globally positive, but #1 and #2 raise a number of important issues. Although I will not dwell into much detail, I would like to highlight the main points.

Reviewer 1 mentions the dilemma that on one hand different conditions to induce inflammation lead to different outcomes and on the other, particular triggers produce similar effects. While s/he is not asking you to solve the conundrum, you should address and discuss it. The reviewer also lists a number of instances where additional and more detailed information and analysis should be provided.

Reviewer 2 is also positive while more reserved. In general, s/he raises a general issue of novelty, which is of great concern for us, and suggests a number of approaches, both experimental and in discussion, to resolve this aspect. Similarly to Reviewer 1, Reviewer 2 also requests a number of

clarifications.

In conclusion, while publication of the manuscript cannot be considered at this stage, given the potential interest of your findings and after internal discussion, we have decided to give you the opportunity to address the above concerns.

We are thus prepared to consider a revised submission, with the understanding that the Reviewers' concerns must be addressed with additional experimental data where appropriate and that acceptance of the manuscript will entail a second round of review. The overall aim is to significantly upgrade the relevance and usefulness of the manuscript.

Please note that it is EMBO Molecular Medicine policy to allow a single round of revision only and that, therefore, acceptance or rejection of the manuscript will depend on the completeness of your responses included in the next, final version of the manuscript.

EMBO Molecular Medicine now requires a complete author checklist (<http://embomolmed.embopress.org/authorguide#editorial3>) to be submitted with all revised manuscripts. Provision of the author checklist is mandatory at revision stage; The checklist is designed to enhance and standardize reporting of key information in research papers and to support reanalysis and repetition of experiments by the community. The list covers key information for figure panels and captions and focuses on statistics, the reporting of reagents, animal models and human subject-derived data, as well as guidance to optimise data accessibility. The checklist will be published with the Peer-Review process file in case of acceptance of your manuscript, in accordance with our Transparent Review Process.

As you know, EMBO Molecular Medicine has a "scooping protection" policy, whereby similar findings that are published by others during review or revision are not a criterion for rejection. However, I do ask you to get in touch with us after three months if you have not completed your revision, to update us on the status. Please also contact us as soon as possible if similar work is published elsewhere.

I look forward to seeing a revised form of your manuscript as soon as possible.

***** Reviewer's comments *****

Referee #1 (Remarks):

In this manuscript the authors re-analyzed the dataset of Seok et al (PNAS 2013) that has resulted in questioning in more general terms the relevance of mouse models to mimic disease conditions in human. This same dataset has been re-analyzed recently also by Takao and Miyakawa (2015) leading to a different conclusion. The diverse outcomes and conclusions drawn raise the question where the problem lies. Are the models poorly reproducing the phenotypes seen in man, have we not yet used the right tools to determine what that right models are, or do we not apply the right analytical tools to analyze them?

In the current study this same dataset was analyzed using Gene set enrichment analysis (GSEA) that utilizes the expression data from all the transcripts, independent on their level of expression, belonging to a particular pathway.

In using this strategy the authors found that a subset of the mouse models actually well mimicked the human condition associated with sepsis. The authors conclude from their study that it is important to select the "right" mouse model for a particular disease indication for translational research purposes.

Critique:

Although it is satisfying that by using GSEA one can find quite good correlations between human disease and some of the mouse models the problem remains how then to identify the appropriate mouse model. According to the authors testing a series of mouse models on the basis of GSEA would be the approach. However it becomes problematic if inducers of disease conditions in the mouse would significantly divert from what causes the condition in man. In this case, one would like to understand on the one hand why quite different conditions to induce inflammation give such diverse outcome and on the other hand why particular triggers to induce inflammation show a high degree of similarity. This is especially relevant since the selection of models is in general based on introducing similar genetic defects (e.g. in KO or transgenic models) or using the same agents to induce the disease. It would be helpful if the authors more clearly point out this dilemma and discuss their observations in this context.

The authors mention thousands of genes being used in their analysis as to not discard information that can be acquired from relatively small differences. They should indicate which gene sets they have actually used for their calculations. Are these the genes listed in Table 1? Is the selection of genes as belonging to a "particular pathway" robust enough? Most pathways are only superficially known whereas especially in case of comparing different species the unidentified components could well play a prominent role.

Some sentences need revision. E.g. In the paragraph "the Paper explained" they cite two groups but with the same sentence "genomic responses in mouse models poorly mimic human inflammatory diseases". I presume that is not what they meant. It also helps the review process when page numbers are indicated. CD14 and CD41 are both used for denoting CD14. There are additional sentences that do not read properly.

It would be helpful if the authors included an enrichment plot with normalized enrichment score and q values for false discovery rates of expression levels of genes acting in the Toll receptor pathway for both human disease and mouse models.

Similarly, it would be useful to explore the similarities and dissimilarities between samples: a sample to sample heat map or sample to sample distance matrix for upregulated and downregulated genes in the inflammatory pathways in both mouse and man.

Referee #2 (Remarks):

Weider and collaborators tackle the question of the relevance of biological data in different mouse models to mimic and study human inflammatory diseases. This question has been addressed in two previous articles (Seok et al, 2013 and Takao&Miyakawa, 2015) using the same data resulting in contradictory results. In contrast to previous approaches, the authors follow a strategy that does not restrict the analysis to genes that are highly up or downregulated in the disease samples. Instead, they performed a gene set enrichment analysis that compares the transcriptional modulation of predefined gene sets of pathways that are assigned to inflammatory processes in either human or mouse. This approach has the advantage of analyzing a whole set of related genes rather than selected subset of genes, and focus on the analysis of biological pathways and not individual genes.

The authors claim that this approach is fundamentally different from assigning GO terms or pathways to genes after filtering for strongly regulated genes. I only partially agree with the authors, as a pathway where most of their gene members are strongly up or downregulated should be captured following both a gene-based approach and a GSEA method. Accordingly, one would expect an overlap on the pathways significantly enriched detected by both methods. In this context, I

miss a detail comparison of the significantly regulated pathways identified in this study and those found by Takao and Miyakawa,2015 using a gene based approach. In this comparison, the authors should pay particular attention to the gain in biological information that their approach provides compared with the Takao and Miyakawa,2015, in terms of the enriched pathways that both methods found.

In general it is unclear what is the novelty that this analysis provide compared to the study of Takao and Miyakawa,2015. I suggest the authors to highlight in the manuscript the distinct contribution of this analysis to the subject. It seems that the method is able to point to the mouse model that better mimics the human disease. However, cannot the gene-based approach proposed by Takao and Miyakawa,2015 do the same?. It would be interesting to compare their results to the correlations between mouse-human samples of Takao and Miyakawa,2015 and analyze if the results are qualitatively different.

I find the paragraph describing the results of the GSEA a bit confusing. This paragraph does not reflect the large differences between human-human, mouse-mouse and human-mouse comparisons that the authors claim to observe in the manuscript (they say that human datasets correlate with each other very well and that various distinct mouse datasets only showed a slight correlation). Human datasets have average positive and negative predictive values of 61%. The comparison between mouse datasets reveals an average positive and negative predictive value of 44%. The authors add that "strikingly, the overlap between mouse and human revealed average positive and negative predicted value of 48% for all human and mouse datasets". The numbers of the human-human comparison (61%) are just slightly higher than those of human-mouse comparison (48%) and the latter is even higher than the mouse-mouse comparison (44%). These numbers show that similar differences in pathway regulation are found within and across species, and suggest that these differences are due to differences in samples rather than a species differences. There is not evaluation on how significant are the differences they observe. I would suggest to do this in order to substantiate their claims.

Referee #3 (Remarks):

Albeit I am not a statistician, I really enjoyed and followed the points made by Dr. Weidner and colleagues. In their paper, the authors compared transcriptomic datasets between human/human, mouse/mouse and mouse/human addressing the IMPORTANT question if data obtained in pre-clinical (mouse) models can be translated in the human setting. Their main finding is a moderate overlap between mouse/mouse and human/mouse data for most datasets. Interestingly, in some diseases a high correlation was observed.

Overall, this paper adds to setting the value of preclinical animal models, which need to be determined for each model individually.

Article EMM-2015-06025: Point-by-Point Response to Reviewers' Comments

Referee #1 (Remarks):

In this manuscript the authors re-analyzed the dataset of Seok et al (PNAS 2013) that has resulted in questioning in more general terms the relevance of mouse models to mimic disease conditions in human. This same dataset has been re-analyzed recently also by Takao and Miyakawa (2015) leading to a different conclusion. The diverse outcomes and conclusions drawn raise the question where the problem lies. Are the models poorly reproducing the phenotypes seen in man, have we not yet used the right tools to determine what that right models are, or do we not apply the right analytical tools to analyze them?

In the current study this same dataset was analyzed using Gene set enrichment analysis (GSEA) that utilizes the expression data from all the transcripts, independent on their level of expression, belonging to a particular pathway. In using this strategy the authors found that a subset of the mouse models actually well mimicked the human condition associated with sepsis. The authors conclude from their study that it is important to select the "right" mouse model for a particular disease indication for translational research purposes.

Critique:

Although it is satisfying that by using GSEA one can find quite good correlations between human disease and some of the mouse models the problem remains how then to identify the appropriate mouse model. According to the authors testing a series of mouse models on the basis of GSEA would be the approach. However it becomes problematic if inducers of disease conditions in the mouse would significantly divert from what causes the condition in man. In this case, one would like to understand on the one hand why quite different conditions to induce inflammation give such diverse outcome and on the other hand why particular triggers to induce inflammation show a high degree of similarity. This is especially relevant since the selection of models is in general based on introducing similar genetic defects (e.g. in KO or transgenic models) or using the same agents to induce the disease. It would be helpful if the authors more clearly point out this dilemma and discuss their observations in this context.

Authors reply:

We fully agree with the reviewer in that establishing animal models to understand human disease is a complicated task and a challenge for most of translational science. Indeed, more and more reports are being published that discuss why new therapies or interventions shown to be effective in animal studies are often less effective or ineffective in clinical trials (Hooijmans & Ritskes-Hoitinga, 2013, see the full reference below). In this context, there are several reasons discussed why animal models of disease can/can't be reliably translated into human (van der Worp, Howells et al., 2010). Albeit we cannot solve this dilemma, we added a discussion in the according paragraph and also more clearly discussed the benefits and limitations of our method in this context (pg. 9).

We would like to emphasize that our GSEA approach will not overcome some basic difficulties in developing the appropriate animal model before any (transcript)omics data of this model have been generated. Mostly, similar agents are used in both human and mice to induce diseases with similar phenotypes. The same holds true for the introduction of genetic defects that target species homologous. This does not reflect the possible variety of outcomes resulting from the diversity of biological processes across species barriers (like mechanisms of adaptation or different pharmacokinetics upon gene deletion or treatment with similar triggers/inducers). Importantly, GSEA is not a tool to predict how to design new model systems but is an effective tool to decide how to interpret existing data in a standardized way, which may add value to the careful selection of the right animal model, thus avoiding unnecessary and misleading translational studies.

References:

- Hooijmans CR, Ritskes-Hoitinga M (2013) Progress in using systematic reviews of animal studies to improve translational research. *PLoS Med* 10: e1001482
- van der Worp HB, Howells DW, Sena ES, Porritt MJ, Rewell S, O'Collins V, Macleod MR (2010) Can animal models of disease reliably inform human studies? *PLoS Med* 7: e1000245

Critique:

The authors mention thousands of genes being used in their analysis as to not discard information that can be acquired from relatively small differences. They should indicate which gene sets they have actually used for their calculations. Are these the genes listed in Table 1?

Authors reply:

We attached the gene sets used for the GSEA analyses as Expanded View Dataset EV1 named 'Gene_sets_Inflammation_BIOCARTA_KEGG_REACTOME.gmt'. Table 1 only lists genes that are involved in the Toll receptor cascade (Reactome) pathway and that were upregulated in at least 9 of 11 datasets. Table 1 shows one result from the GSEA and comprises one of the most commonly regulated inflammatory pathways.

Critique:

Is the selection of genes as belonging to a "particular pathway" robust enough? Most pathways are only superficially known whereas especially in case of comparing different species the unidentified components could well play a prominent role.

Authors reply:

We agree with the reviewer that for many signaling pathways, especially in the field of immunology, probably not all important genes and their functions are identified so far. Thus, our biological understanding of how those pathways are exactly regulated is subject to constant improvements. However, assigning single genes to groups of similar biological functions, signaling processes and pathways is a valuable strategy to objectively include biological knowledge into the data analyses. GSEA makes use of a gene set (=pathway) permutation in order to determine the statistical significance for the measured pathway. We used 1000 permutations to create random pathways that are used as background against the defined inflammatory pathways. This strategy ensures a robust measurement for the pathway, albeit future knowledge acquisition in particular pathways will improve the GSEA approach.

Of course, comparing pathway regulation between different species can be prone to errors due to species-dependent genome constitution and regulation mechanisms, raising the main question of the current debate: how conserved is the regulation of genes and biological processes between men and mice? We hypothesized that the conservation is higher at pathway level than on single gene level, and used the GSEA approach to identify murine disease models that resemble human disorders at pathway level.

Critique:

Some sentences need revision. E.g. In the paragraph "the Paper explained" they cite two groups but with the same sentence "genomic responses in mouse models poorly mimic human inflammatory diseases". I presume that is not what they meant.

Authors reply:

We apologize for any confusion and corrected the sentence for the second group of authors accordingly ('greatly' instead of 'poorly'; pg. 13).

Critique:

It also helps the review process when page numbers are indicated.

Authors reply:

We inserted page numbers in order to facilitate the review process.

Critique:

CD14 and CD41 are both used for denoting CD14.

Authors reply:

We have corrected this term in the text (pg. 8).

Critique:

There are additional sentences that do not read properly.

Authors reply:

We are grateful for that note and carefully revised the language style of our manuscript.

Critique:

It would be helpful if the authors included an enrichment plot with normalized enrichment score and q values for false discovery rates of expression levels of genes acting in the Toll receptor pathway for both human disease and mouse models.

Authors reply:

For further improving the comprehensibility we added enrichment plots for the Toll-like receptor pathway for both human disease and mouse models in the Appendix Figure S1 of our revised manuscript. The figure shows normalized enrichment scores, nominal (uncorrected) *P* values and false discovery rates.

Critique:

Similarly, it would be useful to explore the similarities and dissimilarities between samples: a sample to sample heat map or sample to sample distance matrix for upregulated and downregulated genes in the inflammatory pathways in both mouse and man.

Authors reply:

We fully recognize the importance of systematically comparing upregulated and downregulated genes between human and mouse, e.g. for defining potential drug targets. However, we clearly have to admit that the identification of congruently or differently regulated single genes is *i)* beyond the scope of our study and *ii)* not possible with the approach we presented here. The aim of our study was to add value to the discussion of how to interpret data in order to identify suited animal models - based on existing data by the use of pathway-derived gene set enrichment analyses. As discussed above this could be helpful e.g. when choosing the suitable genetic background of mice strains.

However, to fulfill the reviewers request, we additionally performed principal component analyses of the inflammatory gene expression profiles to properly answer this question (see Figure Ref#1_1A below, which is not supposed to be included in the supplement of the revised manuscript). Thus, we could identify several genes that were congruently and differently regulated in mouse and men, respectively. Genes that were induced throughout all human and mouse datasets included the interleukin 1 receptor type I (IL1R2), the interleukin 1 receptor antagonist (IL1RN), the matrix metalloproteinase 9 (MMP9), the peptidoglycan recognition protein 1 (PGLYRP1) and the suppressor of cytokine signalling 3 (SOCS3) (Fig Ref#1_1B and Table Ref#1_1, which are not supposed to be included in the supplement of the revised manuscript). However, the example of MMP9 demonstrates that promising data from animal models may not be successfully translated into clinical practice. The MMP family has been a pharmaceutical target for a long time, but none of the developed drugs has passed clinical trials so far (Fingleton, 2008, see the full reference below). Nonetheless, the therapeutic potential of the MMP family is still of great interest because of its role in many pathological processes, including inflammation (Vandenbroucke & Libert, 2014). The complexity of MMP functions once again clearly shows that biological processes should not only be investigated at single gene basis. Instead, pathway analyses could be helpful for investigation of complex interactions.

References:

- *Fingleton B (2008) MMPs as therapeutic targets--still a viable option? Semin Cell Dev Biol 19: 61-8*
- *Vandenbroucke RE, Libert C (2014) Is there new hope for therapeutic matrix metalloproteinase inhibition? Nat Rev Drug Discov 13: 904-27*

Figure Ref#1_1. Gene expression analyses on inflammatory genes detected in human and mouse.

Expression analyses for selected studies were performed on expression data of individual genes assigned to inflammatory pathways (BioCarta, Reactome and KEGG databases) used for GSEA. Genes were not filtered regarding expression values or statistics.

- A Principal component analysis on individual inflammatory genes for 7 human (GSE37069, GSE36809, GSE3284, GSE9960, GSE13015, GSE13904, GSE28750) and 4 mouse studies (GSE20524, GSE19668, GSE5663 (CLP), GSE5663 (CLPmild)). Gene groups were manually selected for subsequent inspection (B). Principal component analysis was performed by using Mayday (Battke F, Symons S, Nieselt K (2010) Mayday-integrative analytics for expression data. BMC Bioinformatics 11: 121).
- B Heat maps of genes that were expressed concordantly up (group A), concordantly down (group B) or dissentingly up and/or down (groups C and D) in human and mouse studies. Gene expression data were plotted as log ratio vs. control (blue, decreased; red, increased). Expression of IL1R2, MMP9, PGLYRP1, SOCS3 and IL1RN was concordantly increased throughout all human and mouse data sets (see Table Ref#1_1 below).

Table Ref#1_1

Expression data of genes concordantly induced in inflammatory studies in both human and mouse.

Gene Symbol	Human															
	GSE37069		GSE36809		GSE3284		GSE9960		GSE13015		GSE13904		GSE28750		GSE6269	
	Ratio	P value	Ratio	P value	Ratio	P value	Ratio	P value	Ratio	P value	Ratio	P value	Ratio	P value	Ratio	P value
IL1R2	5.4	0.0000	2.9	0.0000	9.0	0.0076	1.4	0.1884	10.7	0.0000	18.5	0.0000	14.1	0.0000	4.1	0.4288
MMP9	15.5	0.0000	10.7	0.0000	9.1	0.0005	1.8	0.0169	10.7	0.0000	15.0	0.0000	8.1	0.0000	4.9	0.2417
PGLYRP1	14.0	0.0000	12.9	0.0000	5.1	0.0042	2.3	0.0132	6.1	0.0001	5.1	0.0001	5.1	0.0002	45.4	0.3328
SOCS3	3.8	0.0000	2.6	0.0000	7.3	0.0130	1.3	0.2421	5.3	0.0000	5.0	0.0006	4.9	0.0000	2.5	0.0052
IL1RN	2.8	0.0000	2.0	0.0000	22.2	0.0028	1.6	0.0734	3.4	0.0508	2.8	0.0000	1.6	0.0002	2.7	0.0224

Gene Symbol	Mouse																	
	GSE7404(burn)		GSE7404(trauma)		GSE7404(LPS)		GSE5663(LPS)		GSE26472		GSE20524		GSE19668		GSE5663(CLP)		GSE5663(CLPmild)	
	Ratio	P value	Ratio	P value	Ratio	P value	Ratio	P value	Ratio	P value	Ratio	P value	Ratio	P value	Ratio	P value	Ratio	P value
IL1R2	7.0	0.0000	9.2	0.2750	2.5	0.0101	4.0	0.0236	1.9	0.2150	26.5	0.1711	46.9	0.0000	3.8	0.0018	4.1	0.0039
MMP9	2.9	0.0001	2.0	0.2120	1.5	0.1304	1.7	0.1404	1.4	0.5507	5.2	0.0919	6.1	0.0000	2.5	0.0001	2.3	0.0011
PGLYRP1	3.0	0.0012	1.9	0.1792	1.6	0.0977	1.7	0.1530	2.4	0.0954	7.5	0.0875	4.4	0.0003	2.6	0.0100	2.6	0.0073
SOCS3	2.9	0.0016	1.2	0.6350	1.4	0.2287	1.1	0.2718	2.7	0.0683	5.5	0.0014	12.7	0.0013	1.6	0.0000	2.0	0.0506
IL1RN	3.9	0.0022	1.8	0.3942	2.1	0.0226	1.7	0.0007	2.4	0.0711	4.2	0.2952	43.8	0.0015	2.6	0.0014	2.4	0.0001

Gene expression data are presented as linear fold-change ratio over the appropriate control group with respective nominal *P* value. Gene symbols refer to the human gene nomenclature. Genes were selected based on principal component analyses (Fig Ref#1_1, see above).

Referee #2 (Remarks):

Weider and collaborators tackle the question of the relevance of biological data in different mouse models to mimic and study human inflammatory diseases. This question has been addressed in two previous articles (Seok et al, 2013 and Takao&Miyakawa, 2015) using the same data resulting in contradictory results. In contrast to previous approaches, the authors follow a strategy that does not restrict the analysis to genes that are highly up or downregulated in the disease samples. Instead, they performed a gene set enrichment analysis that compares the transcriptional modulation of predefined gene sets of pathways that are assigned to inflammatory processes in either human or mouse. This approach has the advantage of analyzing a whole set of related genes rather than selected subset of genes, and focus on the analysis of biological pathways and not individual genes.

Critique:

The authors claim that this approach is fundamentally different from assigning GO terms or pathways to genes after filtering for strongly regulated genes. I only partially agree with the authors, as a pathway where most of their gene members are strongly up or downregulated should be captured following both a gene-based approach and a GSEA method.

Authors reply:

We agree with the reviewer that for studies, in which the individual gene effect is marked and the variance is small across individuals, single-gene methods are similar powerful as gene set enrichment analyses. However, these prerequisites are not met in many disease states including complex inflammatory disorders (especially for the datasets presented by Seok et al. and Takao & Miyakawa), where single gene effects are often small and more variant across samples. In those cases, GSEA approaches efficiently extract information of the major part of less (but concerted) regulated genes. Therefore, GSEA has become widespread and was proven a very successful approach in many fields of biomedical research over the last decade.

An important advantage of GSEA approaches is that it does not need any biased a-priori filtering of genes that are subjectively defined as important based on individual expression thresholds. Importantly, we should remember that the different setting of gene expression thresholds was the major contributor leading to the contradictory

conclusions drawn from the studies by Seok et al. and Takao & Miyakawa. As presented, GSEA approaches are valuable tools for improving the standardization of bioinformatic analyses of genomic (and related 'omic) studies (discussed in the last paragraph of the our manuscript, pg. 7).

Critique:

Accordingly, one would expect an overlap on the pathways significantly enriched detected by both methods. In this context, I miss a detail comparison of the significantly regulated pathways identified in this study and those found by Takao and Miyakawa, 2015 using a gene based approach. In this comparison, the authors should pay particular attention to the gain in biological information that their approach provides compared with the Takao and Miyakawa,2015, in terms of the enriched pathways that both methods found.

Authors reply:

We kindly acknowledge the view of the Reviewer. However, we only partly agree that a significant overlap of the pathways analyzed by single-gene-based approach and GSEA would only be expected for studies where the individual gene effect is marked and the variance across individual samples is small. In contrast, for other studies in which the individual gene effect is low and the variance across individual samples is high, the major part of less significantly regulated genes can have a considerable effect as discussed in (Mootha, Lindgren et al., 2003, see the full reference below).

Given the high number of significant pathways (several thousand!) reported by Takao and Miyakawa (Takao and Miyakawa, Dataset S1) and the usage of different pathway databases, a detailed comparison of the pathways is beyond the scope of our report. An additional limitation is their restriction to only a few conditions that were apparently chosen in order to directly compare similar stimuli (e.g. burn vs. burn, trauma vs. trauma etc.). This traditional type of pathway analysis (working with gene lists for strongly up- and downregulated genes) leads to limitations in the interpretation of the data: When reporting pathway enrichment, the restriction on P values only considers the statistical significance for the pathway, but not the magnitude of that change (this is because individual gene expression values were omitted before this analysis). In addition, this type of pathway analysis performs a separated calculation for supposedly significantly up- and downregulated genes leading to confusing interpretation. For example, *lymphocyte differentiation* (Takao

and Miyakawa, Fig 4 C) appears to be *simultaneously* activated and suppressed in human burn and trauma. The same is true for *genes involved in cytokine signaling in immune system* (Takao & Miyakawa, Fig 4 B) in the human sepsis study, which shows a significant activation with the GSEA approach (our revised manuscript, Fig 2, dataset GSE28750, line 17). In general, a detailed comparison between our results and that of Takao and Miyakawa is not possible due to missing information concerning their data handling (e.g. it is unclear what time points were used for the human studies).

Reference:

- Mootha VK, Lindgren CM, Eriksson KF, Subramanian A, Sihag S, Lehar J, Puigserver P, Carlsson E, Ridderstrale M, Laurila E, Houstis N, Daly MJ, Patterson N, Mesirov JP, Golub TR, Tamayo P, Spiegelman B, Lander ES, Hirschhorn JN, Altshuler D et al. (2003) PGC-1 α -responsive genes involved in oxidative phosphorylation are coordinately downregulated in human diabetes. *Nat Genet* 34: 267-73

Critique:

In general it is unclear what is the novelty that this analysis provide compared to the study of Takao and Miyakawa,2015. I suggest the authors to highlight in the manuscript the distinct contribution of this analysis to the subject. It seems that the method is able to point to the mouse model that better mimics the human disease. However, cannot the gene-based approach proposed by Takao and Miyakawa,2015 do the same?. It would be interesting to compare their results to the correlations between mouse-human samples of Takao and Miyakawa,2015 and analyze if the results are qualitatively different.

Authors reply:

The novelty of our approach has several key aspects. *First*, by using a GSEA approach instead of single-gene analyses we are able to circumvent any problems associated with subjectively setting of gene expression thresholds that led to opposite conclusions by Seok et al. vs. Takao&Miyakawa, thus allowing us to analyze the datasets in an unbiased, standardized manner. *Second*, we included all datasets presented by Seok et al. (besides the ARDS study due to lack of healthy controls), whereas Takao & Miyakawa only presented analyses for a selection of datasets. *Third*, by using pathway-based GSEA we focused exclusively on genes that were annotated to be involved in inflammatory processes, thus specifically addressing the (patho)physiological process of question. *Forth*, we thus could identify

key pathways that might play dominant roles for the translation of human disease conditions to the mouse model. *Fifth*, our GSEA approach is able to separate mouse models with high predictivity for the human condition from those with low predictivity. Since Takao & Miyakawa only analyzed a selection of the inflammatory datasets presented by Seok et al. (missing studies comprise GSE3284, GSE9960, GSE13015, GSE13904, GSE6269, GSE7404/LPS, GSE5663, GSE26472), we are not able to completely compare our results with those of Takao & Miyakawa. In their study, 3 human (GSE37069, GSE36809 and GSE28750) and 4 murine datasets (GSE7404/burn, GSE7404/trauma, GSE19668 and GSE20524) were correlated to each other. In accordance with our approach, Takao & Miyakawa reported a good correlation for these particular datasets. To enable a global comparison of both data analysis strategies, single-gene-based vs. GSEA approaches, we manually calculated the correlation for the missing datasets based on the single-gene-based approach described by Takao & Miyakawa (albeit it was not completely reproducible due to missing information concerning data handling).

Our results of that single-gene-based correlation are shown in Appendix Figure S2 of our revised manuscript. Although there are some similarities to our results (Fig 1B of our revised manuscript), there are also striking dissimilarities: according to the GSEA approach murine GSE5663 (CLP) and GSE5663 (CLP mild) have a good predictivity with respect to human GSE3284 (LPS), but the rank correlation between the two mouse models and the human model is low when calculated with the single-gene-based approach. On the other hand, the single-gene-based approach yielded numerous studies that showed a medium to good correlation to each other. In contrast, our GSEA approach partly revealed no correlation for these studies (e.g. the murine LPS model GSE7404 against all human studies), making it difficult to identify the optimal mouse model. Apparently, the single-gene approach used by Takao & Miyakawa tends to overestimate the correlation between mouse and human data due to the fact that only overlapping differentially expressed genes were compared to each other, omitting all relevant information of other genes involved in inflammation (as opposed to Seok et al., who included all human genes that were differentially expressed leading to very low correlation to mouse data). Noteworthy, we could not perform a common correction of P values for multiple hypothesis testing (e.g. by Benjamini-Hochberg), because it would lead to some data sets with no differentially expressed genes!

The problem of different data handling strategies is avoided with the GSEA approach, since all genes of interest are considered for analysis, irrespective of their fold change values and statistics. Additionally, our approach to focus on specific biological pathways of interest (relevant for the highlighted diseases) minimizes the background noise of potentially unintended genes.

Indeed, there exists a fundamental qualitative difference between both approaches. When the mouse models are ranked according to the average predictive capability using the GSEA approach on the one hand and single-gene-based approach used by Takao & Miyakawa on the other hand, the correlation between both rankings is low ($r = -0.35$) and statistically not significant ($P = 0.36$, test for correlation between paired samples using Spearman's ρ , applying the R function `cor.test`).

Critique:

I find the paragraph describing the results of the GSEA a bit confusing. This paragraph does not reflect the large differences between human-human, mouse-mouse and human-mouse comparisons that the authors claim to observe in the manuscript (they say that human datasets correlate with each other very well and that various distinct mouse datasets only showed a slight correlation). Human datasets have average positive and negative predictive values of 61%. The comparison between mouse datasets reveals an average positive and negative predictive value of 44%. The authors add that "strikingly, the overlap between mouse and human revealed average positive and negative predicted value of 48% for all human and mouse datasets". The numbers of the human-human comparison (61%) are just slightly higher than those of human-mouse comparison (48%) and the latter is even higher than the mouse-mouse comparison (44%). These numbers show that similar differences in pathway regulation are found within and across species, and suggest that these differences are due to differences in samples rather than a species differences. There is not evaluation on how significant are the differences they observe. I would suggest to do this in order to substantiate their claims.

Authors reply:

We thank the reviewer for raising this very important point and performed a statistical rank sum test with those three comparisons (h-h vs. m-m vs. h-m). We calculated a statistically significant difference ($p < 0.0001$, Kruskal-Wallis test followed by Dunn's multiple comparisons test) between human-human and mouse-mouse correlations

underpinning our claim of interspecies differences. We added this aspect in the main text (pg. 5) and amended Figure 1A of our revised manuscript accordingly (pg. 17). We also would like to emphasize that positive and negative predictive values (which is the part of overlapping pathways) alone do not fully address the question how meaningful the overlap in pathway regulation is, since it could be a matter of chance. Instead, the increase over estimation by chance is a better variable to evaluate the overlap in pathway regulation. As written in the main text, the increase over estimation by chance is +35%, +11% and +19% for all correlations between human-human, mouse-mouse and human-mouse, respectively (pg. 5). To point on this statistical aspect, we also supplemented Fig EV1 (our revised manuscript) that presents the increase of positive and negative predictive values over estimation by chance. Under this aspect, the difference across species (+35% vs. +11%) is much higher. In addition, the number of statistically significant ($p \leq 0.05$) intraspecies correlations of always two datasets was 96% for human which was twice as much as for mouse (47%). Again, this difference was tested to be statistically significant with $P=8.1e-05$ (2-sample test for equality of proportions with continuity correction using the R function `prop.test`).

Referee #3 (Remarks):

Albeit I am not a statistician, I really enjoyed and followed the points made by Dr. Weidner and colleagues. In their paper, the authors compared transcriptomic datasets between human/human, mouse/mouse and mouse/human addressing the IMPORTANT question if data obtained in pre-clinical (mouse) models can be translated in the human setting. Their main finding is a moderate overlap between mouse/mouse and human/mouse data for most datasets. Interestingly, in some diseases a high correlation was observed.

Authors reply:

We thank the reviewer for taking time to review our manuscript.

Thank you for the submission of your revised manuscript to EMBO Molecular Medicine. We have now received the enclosed reports from the referees that were asked to re-assess it. As you will see the reviewers are now globally supportive and I am pleased to inform you that we will be able to accept your manuscript pending final minor amendments

***** Reviewer's comments *****

Referee #1 (Comments on Novelty/Model System):

It is valuable to have different approaches to assess the utility of experimental models of human disease. This manuscript add to the discussion and provides a different, likely more robust and also balanced approach.

Referee #1 (Remarks):

The authors have revised the manuscript and adequately dealt with the criticism of the reviewers.

Referee #2 (Comments on Novelty/Model System):

Statistical methods are properly applied for the different analysis presented.

Referee #2 (Remarks):

The authors have answered all my concerns satisfactorily and in my view, the paper is ready for publication

Corresponding Author Name: Gilbert Schönfelder

Manuscript Number: EMM-2015-06025